# Cross-Sectional Study of Varicella Zoster Virus Immunity in Healthy Korean Children Assessed by Glycoprotein Enzyme-Linked Immunosorbent Assay and Fluorescent Antibody to Membrane Antigen Test

**DOI:** 10.3390/vaccines9050492

**Published:** 2021-05-12

**Authors:** Yunhwa Kim, Ji-Young Hwang, Kyung-Min Lee, Eunsil Lee, Hosun Park

**Affiliations:** 1Department of Microbiology, College of Medicine, Yeungnam University, Daegu 42415, Korea; linbi@ynu.ac.kr (Y.K.); generous81@naver.com (J.-Y.H.); blackxg@naver.com (K.-M.L.); 2Department of Pediatrics, College of Medicine, Yeungnam University, Daegu 42415, Korea; les2055@ynu.ac.kr; 3Immunogenicity Evaluation Laboratory, Clinical Trial Center, Yeungnam University Medical Center, Daegu 42415, Korea

**Keywords:** varicella, vaccine, immunity, glycoprotein, ELISA, FAMA

## Abstract

The prevalence of varicella is especially high among children in the age group of 4–6 years in South Korea, regardless of vaccination. We investigated the immune status of healthy children enrolled in day-care centers and compared pre- and post-vaccination immunity. Antibody titers were measured using a glycoprotein enzyme-linked immunosorbent assay (gpEIA) kit, and the seroconversion rate was assessed using a fluorescent antibody to membrane antigen (FAMA) test. Among 541 vaccinated children, 109 (20.1%) had breakthrough varicella. However, 13 (72.2%) of the 18 unvaccinated children had a history of varicella. The gpEIA geometric mean titers (GMTs) of pre- and 5 weeks post-vaccination in 1-year-old children were 14.7 and 72 mIU/mL, respectively, and the FAMA seroconversion rate was 91.1%. The gpEIA GMTs of 2-, 3-, 4-, 5-, and 6-year-old children were 104.1, 133.8, 223.5, 364.1, and 353.0 mIU/mL, respectively. Even though the gpEIA GMT increased with age, the pattern of gpEIA titer distribution in 4- to 6-year-old vaccinees without varicella history represented both waning immunity and natural boosting immunity. These results suggest that some vaccinees are vulnerable to varicella infection. Therefore, it is necessary to consider a two-dose varicella vaccine regimen in South Korea.

## 1. Introduction

Varicella zoster virus (VZV) causes varicella (chickenpox), which is a highly contagious disease in children with primary infection. The virus persists in individuals during their lifetime and may subsequently reactivate and cause herpes zoster, especially in senescent or immunocompromised individuals [1]. Major symptoms of varicella are fever and rash; however, it sometimes leads to severe complications such as encephalitis, varicella pneumonia, or death [2,3].

A live-attenuated vaccine for varicella has been selectively used in Korea since 1988, and it has been included in the National Immunization Program (NIP) in Korea since 2005 for children aged 12–15 months with a single dose schedule [4]. In Korea, several domestic and imported vaccines have been used, and some vaccines have been discontinued. The varicella vaccine coverage rates for children aged 6 years or less in Korea were 88.3%, 96.8%, and 99.0% in 2007, 2008, and 2009, respectively, and they were over 97% between 2016 and 2019 [5,6,7,8]. Nevertheless, varicella outbreaks occasionally occur in day-care centers or primary schools, even among vaccinated children with mild clinical presentation [9]. Until now, only a small number of reports have been published regarding the long-term immunogenicity of Korean children after varicella vaccination [10]. Therefore, it is necessary to evaluate the immune status of the most vulnerable age groups to investigate the basis of varicella outbreaks. 

Currently, an enzyme-linked immunoassay kit using a VZV glycoprotein (gpEIA) (Birmingham, UK), which is calibrated to international units (IUs), is available. Therefore, we investigated gpEIA antibody titers in children and compared the seroconversion rate as evaluated by fluorescent antibody to membrane antigen (FAMA) testing.

## 2. Materials and Methods

### 2.1. Subjects

In a cross-sectional study, 572 children aged 2 to 6 years old were enrolled, and blood samples were collected during annual health examinations at 13 day-care centers in Gyeongsan city from April 2009 to June 2009. For the vaccination study, 45 children aged 1 year old were enrolled in Yeungnam University Hospital (Daegu, Korea) between May 2013 and August 2015, and blood was collected before and 5 ± 1 weeks after varicella vaccination (Suduvax^®^, GreenCross Company, Yongin-si, Korea). Approximately 2–3 mL of blood was collected, and all sera were stored at −70 °C until use. These studies complied with the Institutional Review Board of Yeungnam University Medical Centers (PCR-09-23, YUH-2013-01-330), and written informed consent was obtained from all of the enrolled children’s parents. Thirteen blood samples were excluded, and 604 samples were analyzed in this study.

### 2.2. Glycoprotein Enzyme-Linked Immunoassay (gpEIA)

Antibody titers of all samples were measured using a VaccZyme^TM^ VZV glycoprotein IgG Low EIA kit (detection range 10–810 mIU/mL, Binding site, Birmingham, United Kingdom, UK). Samples beyond the detection limit of the Low kit were re-analyzed using the Screening kit (detection range 0.5–10 IU/mL). These procedures were performed according to the manufacturer’s instructions.

### 2.3. Fluorescent Antibody to Membrane Antigen (FAMA) Test

FAMA tests were performed using 2-fold serially diluted sera as previously described [11,12]. Phosphate-buffered saline-diluted WHO international standard for VZV immunoglobulin (NIBSC W1044, UK) to 7.8 mIU/mL was used as a positive cut-off reference serum. Cells presenting archetypal membrane fluorescence were regarded as positive using an Axioscope fluorescence microscope equipped with an HBO 50 mercury lamp (Carl Zeiss, Oberkochen, Germany). FAMA titers > 1:16 were considered to be positive.

### 2.4. Statistical Analysis 

X^2^ tests were used to examine statistical differences between age groups, and paired *t*-tests were used for the differences before and after vaccination with varicella. Age-associated increases in geometric mean titer (GMT) changes were assessed by one-way ANOVA (SPSS V. 25.0, IBM SPSS, Inc., Chicago, IL, USA). The Kruskal–Wallis test using Tukey’s multiple comparisons test were used to determine the waning of GMT depending on the duration of the onset of varicella. All tests were performed at a two-sided significance level of *p* < 0.05 (SPSS V. 25.0, IBM SPSS, Inc., Chicago, IL, USA, GraphPad Prism V. 9.0.1., GraphPad Software Inc., San Diego, CA, USA).

## 3. Results

### 3.1. Characteristics of Enrolled Children 

For the cross-sectional study of healthy children, a total of 572 children aged 2–6 years were enrolled in this investigation; however, 559 children were included in the final data analysis because of insufficient information regarding 13 children. In all, 66, 133, 158, 149, and 53 children aged 2, 3, 4, 5, and 6 years, respectively, were included (Table 1). Based on a parent questionnaire, 541 children (96.8%) had been vaccinated, and 18 (3.2%) were not vaccinated. Vaccinations involved four different vaccines; SuduVax (62.2%), Varilrix (10.7%) (GSK, Brentford, UK), Vari-L (1.1%) (Changchun Institute of Biological Products, Changchun-si, China), CJ Sudu vaccine (0.2%) (Cheil Jedang, Seoul, South Korea), and unknown vaccines accounted for 25.7% of the cases. Among the vaccinees, 432 (79.9%) had no history of varicella, and 109 (20.1%) had breakthrough varicella disease. The percentage of children with varicella history among vaccinees were significantly increased from 9.1, 8.5, 20.3, 33.1, to 28.0% in age 2, 3, 4, 5, and 6 years old, respectively (Table 1) (X^2^ test for trend, *p* < 0.0001). Among children who had not been vaccinated, 5 (27.8%) had no history of varicella, and 13 (72.2%) had experienced varicella infection (Table 1). For the vaccine study, 45 one-year-old children with no history of varicella were enrolled. Therefore, in all, 604 children were included in the analysis. Among them, 312 (51.7%) were male and 292 (48.3%) were female (Table 2).

### 3.2. gpEIA Titer and FAMA Seroprevalence of Pre- and Post-Vaccination Sera

The gpEIA GMT of 1-year-old children (*n* = 45) was 14.7 and 72 mIU/mL before and after vaccination, respectively (Table 3) and the gpEIA GMT of post-vaccination was significantly different with pre-vaccination (paired *t*-test, *p* < 0.0001). The range of gpEIA titers was 4.4–53.4 and 10.7–359.1 mIU/mL before and after vaccination, respectively (Table 3, Figure 1a). In pre-vaccination sera, gpEIA titers in 14 sera (31.1%) were lower than 10 mIU/mL, in 17 sera (37.8%) were between 10 and <20 mIU/mL, in 9 sera (20%) were between 20 and <30 mIU/mL, in 2 sera (4.4%) were between 30 and <40 mIU/mL, in 2 sera (4.4%) were between 40 and <50 mIU/mL, and in one serum sample (2.2%), the titer was 53.4 mIU/mL (Figure 1a). In contrast, only 3 sera (6.7%) showed titers lower than 30 mIU/mL, 6 sera (13.3%) showed between 30 and <40 mIU/mL, 5 sera (11.1%) showed between 40 and <50 mIU/mL, 17 sera (37.8%) showed between 50 and <100 mIU/mL, and 14 sera (31.1%) showed titers between 100 and 400 mIU/mL after vaccination (Figure 1a). Therefore, the antibody titer of 88.9% of pre-vaccinated sera was <30 mIU/mL, while 93.3% of post-vaccinated sera showed titers >30 mIU/mL. The percentage of children whose antibody titers were >50 mIU/mL was 2.2% and 68.9% in the pre- and post-vaccination groups, respectively.

All 45 pre-vaccination sera were negative in FAMA testing. The seroconversion rate of post-vaccination sera was 91.1% and the FAMA GMT of post-vaccination was significantly different with pre-vaccination (paired t-test, *p* < 0.0001) (Table 3). FAMA GMT titers were 1.3 and 41.6 in pre- and post-vaccination sera, respectively (Table 3). The range of FAMA titers was <2–4 and 2–512 in pre- and post-vaccination sera, respectively (Figure 1b). Four (8.9%) post-vaccination sera presented negative FAMA results. Among these, the gpEIA titers of three post-vaccination sera were <30 mIU/mL (Table 4). However, the gpEIA titers of subject No. 37, who did not present seroconversion in FAMA test results, were approximately 50 mIU/mL both in pre- and post-vaccination sera, and gpEIA titers were not particularly different after vaccination (Table 4). 

### 3.3. gpEIA GMT of 2- to 6-Year-Old Children

The gpEIA GMT of 2- to 6-year-old children was 215.1 mIU/mL. The gpEIA GMTs of vaccinees who had developed breakthrough varicella (1423.4 mIU/mL) were 10.8-fold higher than those of vaccinees without varicella history (131.4 mIU/mL) (Table 5). In vaccinated children, gpEIA GMTs significantly increased with age regardless of varicella history (*p* < 0.05). Meanwhile, the gpEIA GMTs of children who had neither been vaccinated nor had varicella history were <10 mIU/mL.

### 3.4. Distribution of gpEIA Antibody Titer among Vaccinees without Varicella History

The range of the gpEIA antibody titer of 477 vaccinees without varicella history was very broad, ranging from 1.3 to 51,080 mIU/mL. The distribution of antibody titers showed that the majority of children had titers between 1.3 and 2.4 log_10_ mIU/mL (20.2–310.9 mIU/mL) in all age groups, including 5 weeks after vaccination (Figure 2). The distribution of antibody titers was broadened in >3-year-old children with increasing titers. However, peak incidences were left shifted to 1.3–1.6 log_10_ mIU/mL (20.6–50.0 mIU/mL) in 5- and 6-year-old children compared to 1.7–2.0 log_10_ mIU/mL (51.1–120.4 mIU/mL) at 1 year post-vaccination and 2–4-year-old groups (Figure 2).

### 3.5. Waning of Immunity among Vaccinees with Varicella History

To evaluate the waning of immunity, gpEIA GMT of 109 vaccinees with varicella history was analyzed by the duration between blood collection and the onset of varicella disease. gpEIA GMTs were 3.55 log_10_ mIU/mL within 6 months of the onset of varicella, 2.95 log_10_ mIU/mL between 6 and 18 months, 2.71 log_10_ mIU/mL between 18 and 30 months, and 2.40 log_10_ mIU/mL after 30 months. The significant decreasing tendency in GMT values was evident as time passed until 30 months after the onset of varicella (Kruskal–Wallis test, *p* < 0.01) (Figure 3).

## 4. Discussion

In this study, 2- to 6-year-old healthy Korean children were enrolled to evaluate their varicella immune status. Since the varicella vaccine has been included in the NIP since 2005, only 3.2% of the enrolled children were unvaccinated. The vaccine coverage rate (96.8%) was comparable to the previous studies after 2008 in Korea [5,7]. This rate was about 20% higher than 2001 survey data, which was performed before varicella vaccine was included in NIP [4]. Four hundred two children (74.3%) among total 541 vaccinees were inoculated with two domestic and two imported vaccines, SuduVax (GreenCross Company, Yongin-si, Korea), CJ Suduvaccine (Cheil Jedang, Seoul, Korea), Varilrix (GSK, Brentford, UK), or Vari-L (Changchun Institute of Biological Products, Changchun-si, China). However, vaccine brands were not confirmed in 139 (25.7%) vaccinees. During 2004 to 2008, three other domestic varicella vaccines such as Boryung Sudu vaccine (Boryung Biopharma, Seoul, Korea), Varicella-Kovax (KOREAVACCINE, Ansan-si, Korea), and Varimmune (Dong-Shin Pharmatheutical, Seoul, Korea) were also used in Korea. SuduVax was made with MAV06 strain, and other vaccines were made with strains originated from vOka strain. SuduVax is most prevalent varicella vaccine in Korea. Contrastingly, CJ Sudu vaccine, Boryung Sudu vaccine, Varicella-Kovax, and Varimmune had been withdrawn. There is not enough efficacy data for all of the varicella vaccines used in Korea, especially in healthy children. In this study, all children who had been neither vaccinated nor had prior varicella history were seronegative and naïve to varicella, except for one child (data not shown). VZV is a highly contagious infectious agent with a basic reproductive number R0 of 10–12; therefore, the prevalence of varicella was significantly higher in unvaccinated children (72.2%) than in vaccinated children (20.1%) (chi-square test, *p* < 0.05). The varicella attack rates in unvaccinated children are 78–82%, 72.5%, 48%, 72.3–80%, and 64.5% in the United States (USA), Spain, Germany, Southern Italy, and Turkey, respectively [13,14,15,16,17,18]. Accordingly, the prevalence of varicella in unvaccinated children in this study (72.2%) was comparable to that in other countries except Germany. Furthermore, the breakthrough rate of varicella in communities with single one-dose vaccination schedules was reported to be 23%, 22.9%, 19%, 12.7–32.1%, and 27.7% in the US, Spain, Germany, Southern Italy, and Turkey, respectively [13,15,16,17,18,19]. Therefore, the breakthrough rate in this study (20.1%) was not particularly different from those reported in other countries. There was no difference in the incidence of varicella according to gender (*p* > 0.05, data not shown).

In this study, all of 45 pre-vaccination sera were FAMA negative, and the FAMA seroconversion rate was 91.1% at 5 weeks after vaccination. Vaccine failure was found in four children (8.9%). In the US, the FAMA seroconversion rate is approximately 80% after one dose of vaccine, and many outbreaks of varicella have been reported [20,21], with previous Korean data for the same being reported as 76.7% [22]. Therefore, the FAMA seroconversion rate in our study was slightly higher than that reported in other studies. VZV is a very labile virus, and it is sensitive to temperature, so the transportation and storage of vaccine should be well-controlled. Cold chain management policy was not mandatory in Korea until the early 2000s. Therefore, there was a possibility of reduced vaccine potency before vaccination, and it would be one of the reasons of vaccine failure. To quantify the amount of antibody in children, gpEIA antibody titers were measured. The gpEIA GMT of pre-vaccination sera was 14.7 mIU/mL (negative level) and gpEIA GMT was increased 4.9-fold (72.0 mIU/mL) 5 weeks after vaccination in 1-year-old children. One Korean varicella seroepidemiological study evaluated using Enzygnost VZV IgG ELISA kits (Siemens Healthcare Diagnostics, Eschborn, Germany) showed that the seroprevalence of 1-year-old children is approximately 80% when the equivocal range of ELISA is included as being considered positive [23]. According to a report from the European seroepidemiology network 2, the interpretation criteria of the Enzygnost VZV IgG ELISA diagnostic kit were as follows: positive (>100 mIU/mL), equivocal (50–100 mIU/mL), or negative (<50 mIU/mL). The VaccZyme^TM^ gpEIA kit is different from the Enzygnost ELISA kit because it uses glycoproteins and not whole VZV antigens for coating antigens. Therefore, it is difficult to compare the two kits. If 50 mIU/mL was applied as the cut-off value for the gpEIA kit, the seroconversion rate as measured by gpEIA was 68.9%, and the sensitivity and specificity were 73.1% and 95.9%, respectively. However, if 30 mIU/mL is applied as the cut-off value, the seroconversion rate was 93.3%, and the sensitivity and specificity were 100% and 95.9%, respectively. Since the FAMA seroconversion rate here was 91.1%, the acceptable gpEIA cut-off value for the varicella vaccine might be between 30 and 50 mIU/mL. Therefore, more large-scale vaccine follow-up studies are necessary to obtain a precise cut-off value for gpEIA kits.

The gpEIA GMT of 2- to 6-year-old vaccinated children without prior varicella history was 215.1 mIU/mL, and the gpEIA GMTs of each age group were gradually increased with age from 104.1 to 352.9 mIU/mL (Table 5). This is considered to be an asymptomatic natural boosting effect. In fact, there were outbreaks in two out of the 13 day-care centers studied, just a few months before blood collection. However, 138 (63.3%) of the 218 vaccinated children in the two day-care centers did not have a history of varicella. The gpEIA GMT of the 138 children (220.8 mIU/mL) was approximately 2-fold higher than that of the vaccinees without varicella history in 11 other day-care centers (103.0 mIU/mL). Post-vaccination gpEIA titers showed only one narrow peak between 1.7 and 2.0 log_10_ mIU/mL (Figure 2, Age 1_Post). However, above 2 years of age, the range of gpEIA antibody titers widened, and a second peak also appeared, and the peak incidence was left-shifted to 1.3–1.6 log_10_ mIU/mL at 5 to 6 years of age (Figure 2). According to previous data using Enzygnost anti-VZV/IgG ELISA kit, seropositive rates were gradually decreased from 1 year (65%) to 4 years (49%) old and then increased after 5 years old [4,24]. So, our data and previous data suggested that some children displayed waning vaccine-induced immunity, while others exhibited an asymptomatic natural boosting effect. 

Vaccinees with varicella history had 10.8-fold higher gpEIA titer (GMT: 1423.4 mIU/mL) than children who had been vaccinated without varicella history (GMT: 131.4 mIU/mL), and this result is also similar to that from a previous study [25]. According to single nucleotide polymorphism analysis, there are 24 vaccine-specific sites in attenuated VZV genomes compared to wild-type VZV, most of them consist of the substitution of an AT to GC pair [26]. Even though it is unclear which vaccine sites are related with virulence or antigenicity, both virulence and antigenicity might be attenuated in vaccine strains. We found an interesting phenomenon in which antibody titers waned rapidly in naturally infected children. The gpEIA GMT was approximately 10-fold lower after 30 months than within 6 months after the onset of varicella (Figure 3). However, this study design was cross-sectional and not a follow-up study; we could not confirm the changes in antibody titer in each individual child.

In the US, a two-dose regimen was recommended by the Centers for Disease Control and Prevention to control breakthrough varicella infections [27]. Varicella vaccine has been used as a routine vaccination in many other countries such as Germany, Australia, Canada, Italy, Spain, Israel, Brazil, and Japan. Among them, Germany, Australia, Italy, Spain, and Japan have implemented NIP protocol of two-dose varicella vaccination [28,29,30,31,32]. However, a single dose regimen has been employed in Korea. It is reported that even though the incidence of varicella in post-vaccination birth cohorts is lower than that in pre-vaccination birth cohorts, varicella prevalence is nevertheless high in 4 to 6-year-old children [33]. Our data also showed that varicella incidences of vaccinees were much higher in 4 (20.3%), 5 (33.1%), and 6 (28%)-year-old children compared to 2 (9.1%) and 3 (8.5%)-year-old children. Therefore, we followed up the incidence of breakthrough varicella in participants by phone survey based on parental responses 7 years later. Only 75 of the 559 children responded to the survey. Of the 75 children, 20 (26.7%) experienced varicella when they were of primary school age. There is one prospective case-control and one retrospective case-control studies to examine varicella vaccine effectiveness in Korea; both studies showed that a one-dose regimen is not enough to protect breakthrough infection, especially after 4 years old [22,34]. According to a case-control study conducted with children over 4 years old in the USA, the vaccine effectiveness of one dose and two doses was 86.0% and 98.3%, respectively [35]. In Hong Kong, the effectiveness of a one-dose vaccine was 69.4%. It increased to 93.4% when the primary dose was given at 12 months and then a second dose was given at around 6 years old [36]. However, they mentioned that the two-dose regimen with a short interval should be considered to reduce varicella outbreaks. Considering all of the above results, it appears to be reasonable to boost immunity by inoculation before 4 years of age. 

This study had certain limitations. First, we enrolled children in only two cities, so the results of this study may not be representative of all South Korean children. Nonetheless, the vaccine coverage rate was comparable with that of other studies [6]. Second, because this was a cross-sectional study, we could not determine changes in immunity in individual children. Therefore, long-term follow-up studies are necessary to understand the dynamics of VZV immunity. Nevertheless, to our knowledge, this study is the first report to evaluate the varicella immune status of healthy children aged 1–6 years in Korea and to represent the antibody titer of varicella vaccines as an international unit. 

## 5. Conclusions

A one-dose regimen of varicella vaccine reduced the overall incidence of varicella in healthy Korean children compared to unvaccinated children. However, similar to waning of natural immunity, vaccine-induced immunity seemed to be waning especially after 4 years old. Thus, some of the healthy children are still vulnerable to varicella. Many countries adopt a two-dose regimen for varicella vaccine. This is more effective than a one-dose regimen to protect breakthrough infection and outbreaks. Therefore, a two-dose regimen would be reasonable to protect children from varicella disease.

## Figures and Tables

**Figure 1 vaccines-09-00492-f001:**
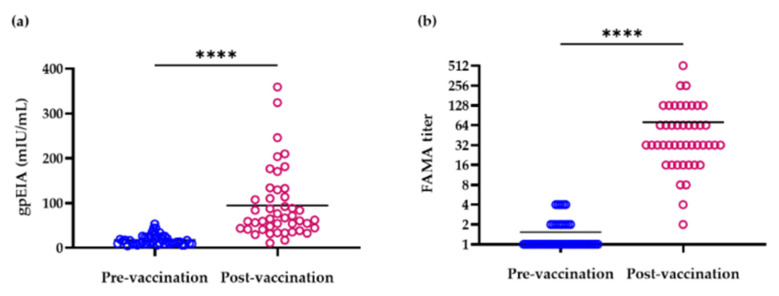
Distribution of gpEIA antibody titers (**a**) and FAMA titers (**b**) in pre- and post-vaccination paired sera. **** *p* value < 0.0001, bar represents mean.

**Figure 2 vaccines-09-00492-f002:**
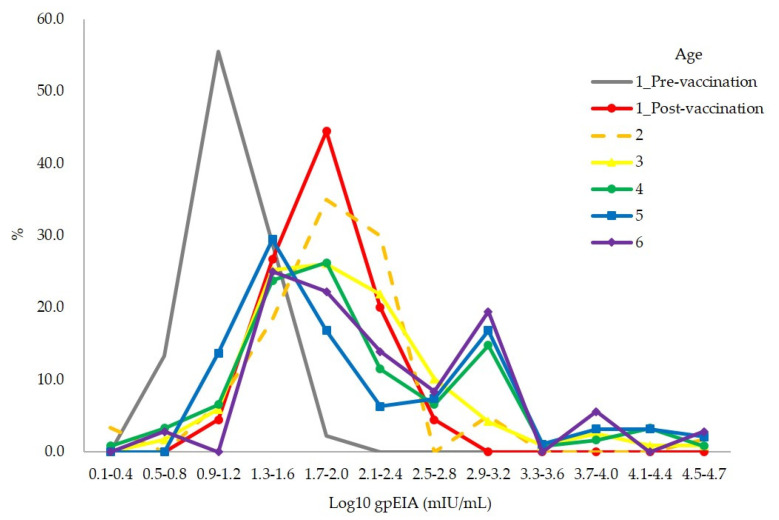
Distribution of gpEIA antibody titers in 477 children who were vaccinated without varicella history.

**Figure 3 vaccines-09-00492-f003:**
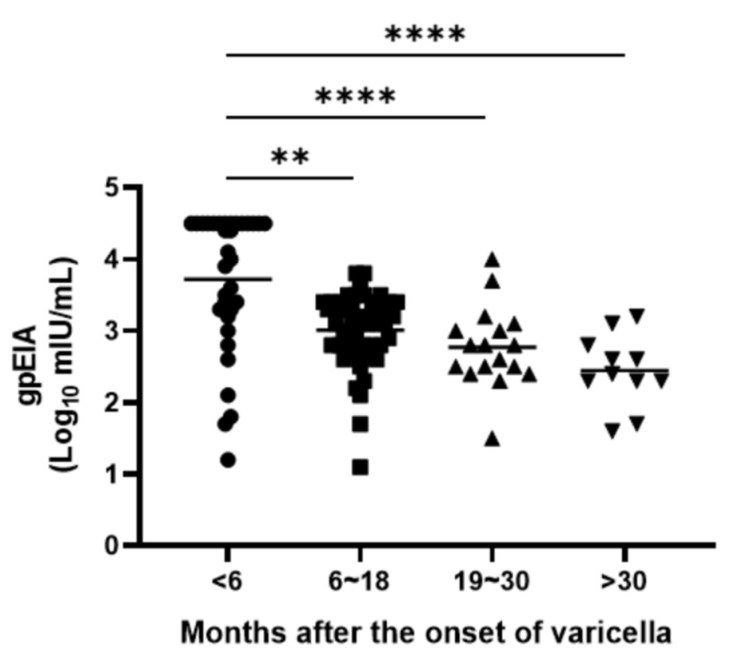
gpEIA titers of 109 vaccinees with varicella history according to the time elapsed from varicella disease. The *p* values were calculated using Kruskal–Wallis test using Tukey’s multiple comparisons test. ** *p* < 0.01, **** *p* < 0.0001.

**Table 1 vaccines-09-00492-t001:** Characteristics of children enrolled in the cross-sectional study.

Age (yr)	Vaccinated (*n* = 541)		Unvaccinated (*n* = 18)	
	Varicella Hx		Varicella Hx	
	−	+	−	+
2	60	6	0	0
3	119	11	0	3
4	122	31	0	5
5	95	47	3	4
6	36	14	2	1
Subtotal (%)	432 (79.9)	109 (20.1)	5 (27.8)	13 (72.2)

Hx; history.

**Table 2 vaccines-09-00492-t002:** Gender distribution of children enrolled in the cross-sectional and vaccine studies.

Age (yr)	Total (*n* = 604)	
	Male	Female
1	20	25
2	29	37
3	66	67
4	81	77
5	92	57
6	24	29
Subtotal (%)	312 (51.7)	292 (48.3)

**Table 3 vaccines-09-00492-t003:** gpEIA and FAMA results of vaccine study paired sera in 1-year-old children.

Measurements	Pre-Vaccination (*n* = 45)	Post-Vaccination (*n* = 45)
gpEIA GMT (mIU/mL)	14.7	72.0 ****
gpEIA titer range (mIU/mL)	4.4–53.4	10.7–359.1
FAMA positive rate (%)	0	91.1
FAMA GMT	1.3	41.6 ****

**** *p* value < 0.0001.

**Table 4 vaccines-09-00492-t004:** Comparison of FAMA and gpEIA titers in subjects who had not seroconverted after vaccination.

Subject No.	FAMA		gpEIA (mIU/mL)	
	Pre-Vaccination	Post-Vaccination	Pre-Vaccination	Post-Vaccination
28	<2	4	6.42	16.96
34	<2	2	8.70	10.65
37	4	8	53.43	59.53
42	2	8	9.53	29.62

**Table 5 vaccines-09-00492-t005:** gpEIA GMT based on vaccination and varicella histories in 2- to 6-year-old children.

Age (yr)	Total (mIU/mL) *	Vaccinated (mIU/mL)		Unvaccinated (mIU/mL)	
		Varicella		Varicella	
		−	+	−	+
2	104.1	85.3	762.7	NA	NA
3	133.8	114.0	571.5	NA	378.2
4	223.5	133.0	1439.3	NA	682.2
5	365.1	167.1	1692.9	9.4	1987.2
6	352.9	220.4	2076.2	7.2	330.0
Total	215.1	131.4	1423.4	8.4	782.3

NA; there were no applicable study subjects, * *p* value < 0.05.

## Data Availability

The data presented in this study are available on request from the corresponding author.

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
