# Peer review of "Cross-Sectional Study of Varicella Zoster Virus Immunity in Healthy Korean Children Assessed by Glycoprotein Enzyme-Linked Immunosorbent Assay and Fluorescent Antibody to Membrane Antigen Test"

_vaccines, 2021, doi:10.3390/vaccines9050492_

Round 1
Reviewer 1 Report
The article “Cross-Sectional Study of Varicella-Zoster Virus Immunity in Healthy Korean Children Assessed by Glycoprotein Enzyme-Linked Immunosorbent Assay and fluorescent Antibody to Membrane Antigen Test” is interesting and suggestive. However, there are several requests and questions to understand the article more clearly.
- The ratio of varicella Hx (+) to (-) increases in 4 to 6-year-old vaccinated children (Table 1). The authors state in the discussion that it appears to be reasonable to boost immunity by inoculation before 4 years of age. I hope that the authors will propose 2-dose varicella vaccination in the abstract in South Koria, if possible.
- The authors should indicate the results of statistical analysis in Table 3 and Figure 1. Is there a significant difference of GMT between pre- and post-vaccination?
- FAMA and gpEIA titers should be arranged in sample number order in Table 4.
- Although *p value; <0.05 is written under Table 5, * is not found in Table 5.
- The value of gpEIA GMT seems to decrease over time in Figure 3. Is there a significant difference of GMT between months after the onset of varicella? The authors should show the significant difference in Figure 3, if so. The vertical axis of Figure 3 is gpEIA GMT or gpEIA?
6.Please check the followings.
(line 105, 109, 113) Fig.1A→Fig.1a (line 111) 17→17 sera (line 120) Fig.1B→Fig.1b
Author Response
Dear reviewer,
Thank you for your constructive criticism. As requested, we have attempted to address each point of your comments to the best of our ability. We have used “yellow highlights” to mark the revisions in the revised file.
1. The ratio of varicella Hx (+) to (-) increases in 4 to 6-year-old vaccinated children (Table 1). The authors state in the discussion that it appears to be reasonable to boost immunity by inoculation before 4 years of age. I hope that the authors will propose 2-dose varicella vaccination in the abstract in South Korea, if possible.
Response: Thank you for your comments. As suggested, we have added the percentage of varicella histories at each age in Results 3.1 (line 97-99) and Discussion (line 274-276). Two doses regimen proposal was also added in abstract (line 27).
2. The authors should indicate the results of statistical analysis in Table 3 and Figure 1. Is there a significant difference of GMT between pre- and post-vaccination?
Response: Thank you for your comments. Yes, both of the gpEIA GMT and FAMA GMT of post-vaccination was significantly increased compared to those of pre-vaccination (paired t-test, p < 0.0001). We have added text in the results (line 109-110, line 124-125) and added statistical significance in Table 3 and Fig. 1. We also changed the scale of Y axis in Fig. 1b.
3. FAMA and gpEIA titers should be arranged in sample number order in Table 4.
Response: Thank you for your comments. As suggested, we have revised the Table 4 in sample number order.
4. Although *p value; <0.05 is written under Table 5, * is not found in Table 5.
Response: Thank you for your comments. As suggested, * has added in Table 5.
5. The value of gpEIA GMT seems to decrease over time in Figure 3. Is there a significant difference of GMT between months after the onset of varicella? The authors should show the significant difference in Figure 3, if so. The vertical axis of Figure 3 is gpEIA GMT or gpEIA?
Response: Thank you for your comments. There were significant differences between < 6 month and other periods. As suggested, we have added statistical differences in test (line 168-170) and Fig. 3. And we also revised vertical axis of Fig. 3 as gpEIA.
6. Please check the followings. (line 105, 109, 113) Fig.1A→Fig.1a (line 111) 17→17 sera (line 120) Fig.1B→Fig.1b
Response: Thank you for your comments. We corrected typos as suggested.
Reviewer 2 Report
They performed a cross-sectional analysis of Varicella-Zoster Immunity in healthy Korean children. They determined the immune status by using glycoprotein enzyme-linked immunosorbent assay (gpEIA) and fluorescent antibody to membrane antigen test (FAMA). They reported that 20.1% of the vaccinated children had breakthrough varicella, and 72.2% of the unvaccinated children had a history of varicella. The vaccination increased the gpEIA mean titers and achieved FAMA seroconversion by 91.1%. Further, they reported that the pattern of gpEIA titer distribution suggested the possible waning immunity. Based on the findings, they proposed to change varicella vaccine regimens from a 1-dose regimen to a 2-dose regimen.
The study is well conducted. The manuscript is well written, and their statement is substantially supported by their findings and interpretation. Although their research topic is based on their domestic health issue, their research will be of wide interest to the readers.
I have some suggestions to improve their manuscript.
#1. They described that the varicella vaccine has been included in the NIP since 2005 and stated the vaccine coverage rate after 2005. I would like to know the vaccine coverage rates before 2005 because this can potentially affect community boosting in Korea.
#2. The possible reasons for the vaccine failure should be discussed, if any. Are there some variations of the efficacy among the 4 vaccines used in Korea? Immune status of the children, e.g., mild immune deficiency?
#3. Based on the data of Figure 2, the authors speculated the waning of the immunity in Discussion. If there are some available reports which show the waning of immunity in a follow-up study, the authors might want to refer to them to strengthen the authors' statement.
#4. The authors referred to a 2-dose protocol in the Discussion. The authors should describe tangible data showing the 2-dose protocol improves the vaccine's efficacy by referring to previous studies.
#5. A conclusion should be included. In the last paragraph, the authors described the limitation. I would suggest preparing the conclusion section after the paragraph. The authors might want to repeat and underscore their statement described at the end of the abstract.
Author Response
Dear reviewer,
Thank you for your constructive criticism. As requested, we have attempted to address each point of your comments to the best of our ability. We have used “yellow highlights” to mark the revisions in the revised file.
#1. They described that the varicella vaccine has been included in the NIP since 2005 and stated the vaccine coverage rate after 2005. I would like to know the vaccine coverage rates before 2005 because this can potentially affect community boosting in Korea.
Response: Thank you for your comments. Before 2005, varicella was not included in the vaccination registration project and there was only one research data. According to the survey conducted in 2001 with 2,800 students at an elementary school in Gyeonggi-do, varicella vaccine coverage rate was estimated approximately 70%. Therefore, it seemed to increase after varicella vaccine inclusion in NIP. We have added text in Discussion (line 180-183).
#2. The possible reasons for the vaccine failure should be discussed, if any. Are there some variations of the efficacy among the 4 vaccines used in Korea? Immune status of the children, e.g., mild immune deficiency?
Response: Thank you for your comments. We have added texts to describe the possible reasons of vaccine failure in discuss (Line 216-219). VZV is a very labile virus, if cold chain management was inappropriate because it was not mandatory in Korea until the early 2000s, it might be one of the possible reasons to reduce vaccine potency and resulted in vaccine failure. Between late 1990s and the early 2000s, 6 domestic varicella vaccines were used in Korea, however, 5 domestic vaccines had been withdrawn. In our study, we could not confirm vaccine brands in 25.7% vaccinees and we could not exclude the possibility of using the withdrawal vaccines. There are not enough data about the efficacy of varicella vaccines used in Korea. We added the tests in Discussion (line 187-195).
#3. Based on the data of Figure 2, the authors speculated the waning of the immunity in Discussion. If there are some available reports which show the waning of immunity in a follow-up study, the authors might want to refer to them to strengthen the authors' statement.
Response: Thank you for your comments. We added following test in Discussion. According to previous data using Enzygnost anti-VZV/IgG ELISA kit, seropositive rates were gradually decreased from 1 year (65%) to 4 years (49%) old and then increased after 5 years old (line 249-251).
#4. The authors referred to a 2-dose protocol in the Discussion. The authors should describe tangible data showing the 2-dose protocol improves the vaccine's efficacy by referring to previous studies.
Response: Thank you for your comments. We added following texts in Discussion. There are one prospective and one retrospective case-control studies to examine varicella vaccine effectiveness in Korea, both studies showed that one dose regimen is not enough to protect breakthrough infection, especially after 4 years old (line 279-282). According to a case-control study conducted with over 4 years old children in USA, vaccine effectiveness of 1 dose and 2 doses were 86.0% and 98.3%, respectively (line 282-284). In Hong Kong, the effectiveness of one dose vaccine was 69.4%. It increased 93.4% when the primary dose was given at 12 month and then second dose was given at around 6 years old. However, they mentioned that two-dose regimen with short interval should be considered to reduce varicella outbreaks (line 284-286).
#5. A conclusion should be included. In the last paragraph, the authors described the limitation. I would suggest preparing the conclusion section after the paragraph. The authors might want to repeat and underscore their statement described at the end of the abstract.
Response: Thank you for your comments. As suggested, conclusion have added (line 299-306).
Round 2
Reviewer 2 Report
The manuscript has been greatly improved by their intensive revision.
I have no additional comment.